

# Rare and different: Anomaly scores from a combination of likelihood and out-of-distribution models to detect new physics at the LHC

**Sascha Caron[1,2], Luc Hendriks[1⋆] and Rob Verheyen[3]**

**1** High Energy Physics, IMAPP, Radboud University Nijmegen,
Heyendaalseweg 135, 6525 AJ, Nijmegen, NL
**2** Nikhef, 1098 XG Amsterdam, Netherlands
**3** Department of Physics and Astronomy, University College London,
Gower St., Bloomsbury, London WC1E 6BT, UK

⋆ luchendriks@gmail.com

## Abstract

We propose a new method to define anomaly scores and apply this to particle physics collider events. Anomalies can be either *rare*, meaning that these events are a minority in the normal dataset, or *different*, meaning they have values that are not inside the dataset. We quantify these two properties using an ensemble of One-Class Deep Support Vector Data Description models, which quantifies *different*ness, and an autoregressive flow model, which quantifies *rare*ness. These two parameters are then combined into a single anomaly score using different combination algorithms. We train the models using a dataset containing only simulated collisions from the Standard Model of particle physics and test it using various hypothetical signals in four different channels and a secret dataset where the signals are unknown to us. The anomaly detection method described here has been evaluated in a summary paper where it performed very well compared to a large number of other methods. The method is simple to implement and is applicable to other datasets in other fields as well.

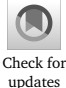

## Contents


## 1 Introduction

One of the currently important challenges in High Energy Physics (HEP) is to find (very) rare new physics signals among the Standard Model (SM) background collision events. An increasingly popular method for finding such anomalous signals is to use anomaly detection techniques derived from Deep Learning (DL), see e.g. [1–20].

The method we propose in this work assigns an anomaly score to each event, that indicates if the event is part of the non-anomalous (SM) dataset. A high anomaly score can be an indicator of physics beyond the Standard Model (BSM), even though the algorithm is not trained on any particular BSM model. In our setup we follow the challenge of [21], where the goal is to find an algorithm that can identify several BSM signals in the SM background. In the challenge, various methods are evaluated on a secret dataset, whose content was unknown to the developers.

The goal of the challenge is to provide an anomaly score to each event, and using the background efficiency of our method we define a threshold value for the anomaly score to separate signal from background. In the signal region (where events have an anomaly score higher than the threshold) one can perform a statistical test to determine whether the data contains an excess of anomalous events.

In contrast to the unsupervised event classification methods proposed in this work, another type of anomaly may arise purely in the form of an overdensity in the bulk of the phase space that departs from the background model. Methods that search for such anomalies were explored in the LHC Olympics [20], and typically involve some form of DL-driven background estimation, as well as a density comparison between the data and the background to identify regions of interest. The method proposed here does not attempt to perform a density comparison, but instead aims to classify singular events as regular or anomalous with an *anomaly score*. With this information, a signal region can be defined around concentrations of anomalous events by analyzing (or selecting events) based on this anomaly score. Then, a counting experiment or template fit can be performed that compares the number of observed events to the expected number of background events, in which systematic uncertainties must be thoroughly accounted for. This approach is in fact identical to the widely-used more traditional techniques, e.g. [22], where boosted decision trees (BDT) are instead used to define a signal score (the BDT output), similar to the anomaly score, and events are selected based on that. Our method defines the signal region differently, but the other steps of the statistical tests, determination of backgrounds via control selection and systematic uncertainty evaluations can be retained.

Event-by-event anomaly scoring can be accomplished in multiple ways. One option is to determine a manifold to which the dataset can be mapped. Classification then proceeds by attempting to map unknown events to this manifold, and the mapped distance from it serves as the anomaly score. Another option is to model the probability density function of the dataset, and then to evaluate the likelihood of unknown events. The former method assigns a high anomaly score to events that are *different* from the training set, while the latter selects events

that are *rare*. Our method combines these two approaches to achieve a sufficiently robust algorithm.

### Detecting events that are *different*

The first method is to define a region or manifold that describes the known data (in our case the SM) using some algorithm, and then to apply it to a test dataset that contains both known (SM) and anomalous (BSM) data. The known data should fall (predominantly) inside this manifold, while the anomalous data should fall (predominantly) outside this manifold.

Autoencoders [23] are a popular choice for this task. Input events are encoded into a latent space, from which the decoder reconstructs the original event. During training, the encoder learns the manifold to which all SM data is mapped in the latent space. During testing, the assumption is that SM data will fall inside this manifold, while BSM data will fall outside. Then, the decoder will be able to reconstruct the events inside the manifold well, while reconstructing the events outside the manifold poorly. The reconstruction loss may then be used as the anomaly score.

However, autoencoders may still reconstruct events well when they lie outside of the manifold spanned by the training data. Furthermore, the intrinsic topology of HEP data does not necessarily fit the Euclidean manifold that autoencoders attempt to map it to [24]. This can lead to poor reconstruction of non-anomalous data, and makes the interpretation of an autoencoder-defined anomaly score more complicated.

We note that in the case in which the training data is produced from a simulator, one can achieve perfect accuracy when the simulator can be inverted. Simulators for HEP data map a set of random numbers to a collision event. If the inverse of the simulator is available, a collision event can be mapped through the inverse of the simulator to obtain the corresponding set of random numbers. If this set of numbers is not within the sampling space of the SM, the event can be classified as a new physics event. In this case, the manifold is implicitly defined by the inverse simulator. While recent advances in such approaches exist [25], an application at large scale is as of yet numerically infeasible.

The One-Class Deep Support Vector Data Description (Deep SVDD) approach from [26] defines the manifold before training in the form of a (multidimensional) point, and the neural network is trained to map every input event to this point. The distance to the point is then the anomaly score. This procedure achieves the same goal as the autoencoder without requiring the assumption that the decoder cannot reconstruct events outside the manifold, as there is no decoder.

The Deep SVDD model as defined in [26] maps the input to a scalar. Such a model may be too simplistic, as it might not capture all the required information to determine whether or not an event is anomalous. The model may for instance learn to extract trivial symmetries of the data like rotational invariance or momentum conservation. We find that the model may be improved by using a multidimensional output (a constant output *vector*) or an ensemble of Deep SVDD models with different random initialisations, such that models can optimize into different minima that represent different relations between the data. Unlike in [27] we use the Deep SVDD method with different output dimensionalities and create an ensemble of them to maximise the amount of information we can extract from these models. We test the correlation between different Deep SVDD models to check if combining them indeed adds value.

### Detecting events that are *rare*

Another avenue to robust anomaly detection is to infer the probability density of the known dataset and evaluate the likelihood of every event in the test dataset. When testing, SM events

should generally be assigned a high likelihood while new physics events should correspond with a low likelihood. However, deriving the probability density function of the SM as a whole analytically is infeasible. Instead, variational inference methods may be used to approximate the probability density. Variational autoencoders (VAEs) [28] and $\beta$-VAEs [29] accomplish this by maximizing the evidence lower bound, and as a result enforce a structured latent space. This structure can be transformed into an anomaly score, quantified for instance as the radius from the center of a Gaussian latent space or by using an information density buffer [30]. However, such methods suffer from the likelihood gap due to the intractable nature of the real likelihood, as well as from the assumption that the decoder is unable to reconstruct events outside the manifold spanned by the training set.

In this work we instead use an autoregressive flow [31], a particular implementation of a normalizing flow [32–34], to approximate the SM probability density more directly. Such a model allows for the direct evaluation of the inferred likelihood for any event, which may then be converted to an anomaly score. We then combine multiple Deep SVDD neural networks, trained with different target values and dimensionalities, with an autoregressive flow model using different combination algorithms. The idea is that this achieves "the best of both worlds" of learning the SM manifold and obtaining the likelihood of an event given the SM.

In section 2 the algorithm used for the Deep SVDD models is detailed. In section 3 the autoregressive flow is explained and section 4 shows how the Deep SVDD model and autoregressive flow model are combined. In section 5 we explain the dataset at hand. Lastly, in sections 6, 7, 8 we show the results, conclusions and discussion respectively. Our work can be summarised in the following set of contributions:

- We use an *ensemble* of Deep SVDD networks to determine if an event is *different*;

- We use an autoregressive flow model to determine how *rare* it is for an event to originate from the Standard Model;

- We combine the *rare* and *different* objectives using several strategies to assign robust anomaly scores to HEP events.

In a comparison paper we show that this model can outperform traditional methods, Deep SVDD methods and autoregressive flow models separately and various autoencoder-type models (AE, VAE, $\beta$-VAE and others) [21]. The code used for this project is available at [1].

## 2 Deep SVDD Networks

The Deep SVDD network architecture set out in [26] is a fully connected network which outputs a constant scalar for every input. We change architecture slightly such that it can also output a vector with repeated elements (see figure 1). The loss is defined as

$$s(x) = \left[ O_n^d - \text{Model}(x) \right]^2 , \tag{1}$$

where the model maps the input $x$ to the same tensor shape as the manifold $O$. In our case, $O$ is a vector of identical scalar values, with the subscript $n$ defining the scalar value and superscript $d$ the number of elements in the vector. For example, $O_3^4$ identifies the vector $(3,3,3,3)$. The optimisation of the Deep SVDD model is thus fundamentally very simple: it is a NN that receives some input $x$ and transforms it to some output $O_n^d$.

The original Deep SVDD network architecture as described in [26] only has a single output value which is the mean of the training set on an untrained network. In the problem at hand

---

[1]`https://github.com/l-hendriks/combined-anomaly-detection-code`

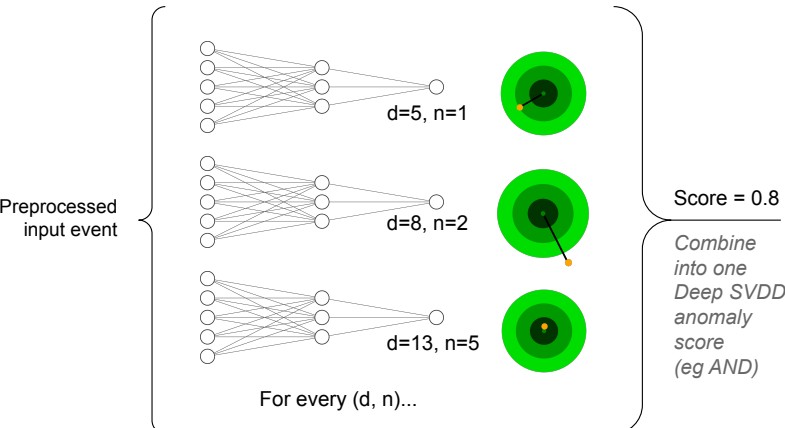

Figure 1: Visualisation of an ensemble of Deep SVDD models. The L2 distance between the network output and the expected fixed number(s) is used as the anomaly score, which is then combined using a combination algorithm described in section 4. The parameters $d$ and $n$ are defined in table 1.

here, many orthogonal relations that could make an event anomalous can be extracted from the data. A single Deep SVDD network with a one-dimensional output will not be able to capture all these relations. We thus employ multidimensional outputs and varying target values and combine an ensemble of these networks using four different combination algorithms described in section 4. We test the correlation between different Deep SVDD models to show the different models actually learn different relations from the data.

All Deep SVDD models have identical network architecture, but different targets. The architecture is a simple dense network with hyperparameters defined in table 2. The activation function of the hidden layers is the exponential linear unit (ELU) [35]. The final layer has $d$ output neurons and a linear activation function. In total 63 models are trained, for all parameter combinations shown in table 1.

Due to its simple nature, the Deep SVDD model is susceptible to finding trivial solutions that contain no physical information. For instance, one such solutions occurs when all network weights vanish, and the biases are equal to the target $n$. In [26] the proposed solution is to set all biases to 0 and to exclude the target value $n = 0$. However, in our experiments, even without these modifications, none of the models were found to converge to a trivial solution, which would be immediately identified as such by evaluating the validation dataset. The likely reason is that the models, which are initialized randomly, are much more disposed to converge to one of many local minima that contain some physical information, instead of to this singular trivial solution.

After combination of all 63 models, a single anomaly score is obtained. Due to the simplicity of the individual models, training a network only takes a few minutes on a NVIDIA GTX 1080Ti graphics card. The hyperparameters used for training are shown in table 2. The

Table 1: Parameter combinations used for training the Deep SVDD models. The hyperparameter $d$ represents the output dimensionality, i.e. the number of output parameters of the network, while $n$ stands for the target output value.

| Hyperparameter | Values |
|---:|:---|
| **d** | 5, 8, 13, 21, 34, 55, 89, 144, 233 |
| **n** | 0, 1, 2, 3, 4, 10, 25 |

Table 2: Parameter combinations used for training the Deep SVDD models.

| Hyperparameter | Value |
|---|---|
| **Initial learning rate** | 0.001 |
| **Batch size** | 10000 |
| **Optimizer** | Adam [36] |
| **Loss function** | Mean squared error |
| **Dense layers** | 3 |
| **Neurons per layer** | [512, 256, 128] |

learning rate is halved when the loss does not improve for 5 epochs and training is stopped when the loss does not improve for 50 epochs.

## 3 Autoregressive flows

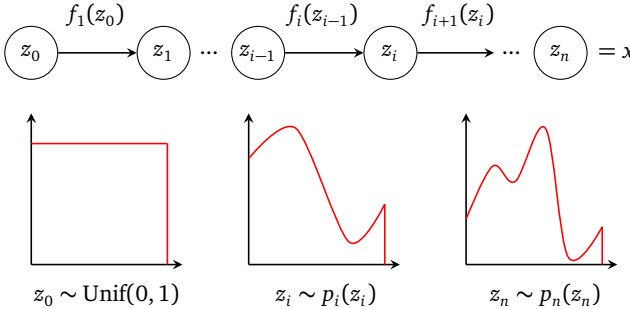

Figure 2: Visualisation of the autoregressive flow. The flow model starts from a uniform distribution and transforms it into a target distribution which is inferred from the training data. After training, the model can be used to evaluate the likelihoods of events.

The autoregressive flow model used here is closely related to the one described in [37], and was specialized in [38] for the purpose of HEP event generation. Here, we provide a short summary of the aspects of the model relevant in the context of anomaly detection.

An autoregressive flow is a probabilistic model that applies a series of bijective and parameterizable variable transforms to a simple prior distribution. Due to the bijective nature of the variable transforms, the model offers tractable likelihood evaluation which may be used during training directly.

In particular, starting from a simple prior distribution $p_0(z_0)$, subsequent latent variables $z_{i+1}$ are determined as

$$z_{i+1} = f_{i+1}(z_i, \theta_i), \tag{2}$$

where $\theta_i$ are parameters inferred during training. The likelihood is then transformed as

$$p_{i+1}(z_{i+1}) = p_i(z_i) \left| \det \frac{\partial z_{i+1}}{\partial z_i} \right|^{-1}. \tag{3}$$

Identifying the last latent dimension $z_n$ with the data $x$, the likelihood may be evaluated by

Table 3: Parameter combinations used for the training of the flow model.

| Hyperparapeter | Value |
|---|---|
| **Initial learning rate** | 0.001 |
| **Batch size** | 512 |
| **Optimizer** | Adam [36] |
| **Loss function** | -LogLikelihood |
| **RQS knots** | 35 |
| **Flow layers** | 11 |
| **MADE layers** | 7 |
| **MADE neurons per layer** | 200 |
| **Epochs (channel 1, 2a, 2b)** | 100 |
| **Epochs (channel 3)** | 10 |

propagating data backwards through the model, such that

$$
\begin{aligned}
\log p(x) &\equiv \log p_n(z_n) \\
&= \log p_0(z_0) + \sum_{i=0}^{n-1} \log \left| \det \frac{\partial z_{i+1}}{\partial z_i} \right|^{-1} .
\end{aligned}
\tag{4}
$$

An illustration of this procedure is shown in figure 2. The transforms $f_i$ should be defined such that the evaluation of equation (4) is efficient. In an autoregressive flow, this is accomplished by factorizing the likelihoods as

$$
p_i(z_i) = \prod_{i=1}^{D} p_i^j(z_i^j | z_i^{1:j}) ,
\tag{5}
$$

where the superscript $j$ indicates the $j$-th feature of the $D$-dimensional data. That is, the likelihood is decomposed in a product of one-dimensional, conditional likelihoods that may be modelled individually. The resulting Jacobian is triangular, allowing for efficient evaluation of the determinant. The corresponding one-dimensional transforms are now

$$
z_{i+1}^j = f_{i+1}^j(z_i^j; \theta_i^j(z_{i+1}^{1:j-1})) ,
\tag{6}
$$

where the parameters $\theta_i^j$ are functionals that may be implemented as deep neural networks. Here, we make use of so-called MADE networks [39], which, together with the choice of eq. (6) enables paralellized inference [31]. Furthermore, the functions $f_i^j$ are chosen to be piecewise Rational Quadratic Splines [37], which are highly expressive and have finite domain. This last property is particularly useful as the phase space of HEP events also has a well-defined boundary.

The autoregressive flow is trained by minimizing the negative log-likelihood over the training data. The hyperparameters are listed in table 3. After training is completed, the anomaly score may be computed as

$$
s(x) = \frac{\log p(x) - \log p_{\min}}{\log p_{\max} - \log p_{\min}} ,
\tag{7}
$$

where $\log p_{\max}$ and $\log p_{\min}$ are respectively the largest and smallest log-likelihoods to appear among the evaluated event samples in the training set. Training the flow model takes around 10 hours on a NVIDIA GTX 1080Ti graphics card.

Table 4: The four algorithms used to combine $N$ anomaly scores $s_i$ into a single anomaly score $S$.

| Name | Combination |
|------|-------------|
| AND | $S(x) = \min(s_i(x))$ |
| OR | $S(x) = \max(s_i(x))$ |
| PROD | $S(x) = \prod_i^N s_i(x)$ |
| AVG | $S(x) = \frac{1}{N} \sum_i^N s_i(x)$ |

# 4 Combining anomaly scores

The scores of the Deep SVDD models are combined into a single score, and that score is again combined with the flow model score, yielding a final anomaly score. We adopt the same four algorithms to combine the scores as in [40], which are quantified in table 4. Before the scores are combined, the training and validation dataset is normalised such that the training set yields a uniformly distributed anomaly score between zero and one. We compute the correlation between the Deep SVDD ensemble and the flow model to show that these methods actually capture different information.

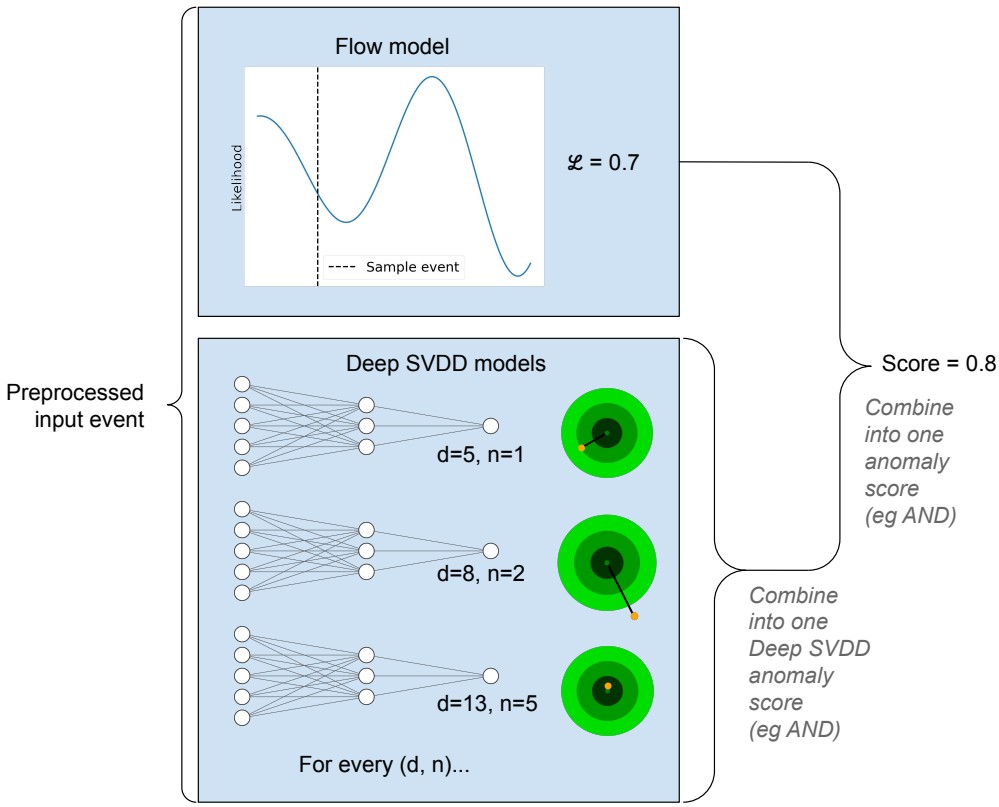

Figure 3: Diagram of the algorithm with example numbers. First, the event is passed through the flow model and Deep SVDD models. From the Deep SVDD models one anomaly score is derived and this score is then combined with the flow model score to obtain a final anomaly score. The combination algorithms are shown in table 4 and all possible values for $d$ and $n$ in table 1.

# 5 Dataset

The dataset that is used to assess the performance of our model was compiled by the Dark-machines collaboration for unsupervised anomaly detection in HEP, and is described in [41]. It contains simulated collider events on the four-vector level. Every event contains the missing transverse energy (MET), missing energy azimuthal angle (MET$\phi$) and, for every object, the object type (which can be a photon, electron, positron, muon, antimuon, jet or b-jet), transverse momentum $p_T$, pseudorapidity $\eta$, azimuthal angle $\phi$ and energy $E$. There are four datasets in total, which are called channel 1, 2a, 2b and 3 respectively. Each channel contains its own training and test set. The training set contains only background data, while the test set contains both background and signal data. The models proposed here were trained on every channel and their performance was assessed on every signal. In total there are 34 different signals, including for instance Z' models and many different supersymmetric models. For a full list, see Table 3 in [21]. In addition to the training set and test set (collectively called the hackathon dataset), a further secret dataset is available, which contains a number of unknown signals. Our algorithm is also tested on this secret dataset to obtain a fair comparison with other models.

Since not every event has the same number of objects, and in particular the autoregressive flow model does not handle categorical data well, the events are transformed to a data structure that fits the method, as follows:

- For the Deep SVDD model, which quantifies how *different* events are, the maximum number of every object type is calculated and the objects in every event are sorted by object type. The data is split into class data and continuous data. The class data is stored as a vector of ones and zeros that indicate whether or not an object is in the event. For example, if the maximum number of jets in the dataset is 4 and in an event there are only 2, the location in the class vector corresponding to jet 1 and 2 are one while jet 3 and 4 are zero. The continuous data follows the same ordering and thus has zeros in positions absent of an object. The continuous data also contains the MET and MET$\phi$. The final input vector is the concatenation of the class and continuous data.

- For the autoregressive flow, which quantifies how *rare* events are, the dataset contains MET, MET$\phi$, the number of jets, b-jets, photons, electrons, positrons, muons, and antimuons, and for the first 7 jets or b-jets and the first 4 leptons (ordered by $p_T$) the object type, E, $p_T$, $\eta$ and $\phi$. The autoregressive flow does not model discrete data well, so the class data is transformed to a uniform random number that is sampled between $\pm 0.5$ from the class value (e.g. if a positron has value 1, it is transformed to a uniformly sampled number between 0.5 and 1.5).

# 6 Anomaly detection results

For every model, channel and signal combination the performance is evaluated using four different metrics: The area under the curve (AUC) and the signal efficiency at three different background efficiencies. The signal efficiency at a particular background efficiency $\epsilon_S(\epsilon_B = x)$ is defined as the true positive rate $\epsilon_S$ when events are cut until the false positive rate is less than a value $\epsilon_B$. We calculate the signal efficiency at three different background efficiencies $\epsilon_B = 10^{-2}$, $\epsilon_B = 10^{-3}$ and $\epsilon_B = 10^{-4}$. These metrics are introduced because the AUC calculates the integrated performance of a model over all background efficiencies, and as such it is dominated by very high background efficiency rates when compared to the necessary background cuts to make new physics signals visible.

To quantify the correlation between models we compute the Pearson Correlation Coefficient, which quantifies the correlation between two sets of data points as the ratio of the covariance of the dataset and the product of the standard deviations of the datasets.

The correlation coefficient for the Deep SVDD models typically range between 0.4 and 0.6 while the correlation between the Deep SVDD ensemble and the flow model is only 0.122. The moderate value of the Pearson coefficient between Deep SVDD models suggests that an ensemble of models should outperform a single one. The correlation between the ensemble of Deep SVDD models and the flow model is very low, suggesting they can be also be combined into a model that outperforms the two separate models.

To assess the overall performance of the different methods, we follow the methodology of [21]. We first calculate the Signal Improvement (SI) of the different methods on the different signals, $\text{SI} = \max\left(\epsilon_S/\sqrt{\epsilon_B}\right)$, for $\epsilon_B = 10^{-2}$, $\epsilon_B = 10^{-3}$ and $\epsilon_B = 10^{-4}$. The SI is a metric that quantifies how well a particular method is in identifying a particular new physics signal when applied on different background efficiency cuts.

To obtain an improvement score over all different signals, the SI values per signal and channel are combined into a Total Improvement (TI) score. In short, the TI quantifies the overall quality of a method over all different signals, channels and background cuts. We combine the SI values into the TI score in three ways: using the minimum, median and maximum of the SI scores. A comparison of the median TI on the hackathon test set and the secret set is shown in figure 4. The results on the hackathon test set and the secret set are shown in figure 5. The performance of the flow model is worse on the secret set than on the hackathon test set, while the combined models do better. This is likely because there are more *different* events in the secret dataset than in the hackathon dataset.

On the hackathon testset, the flow model has the highest minimum and median TI, but the maximum TI is lower than that of the combined model. The individual Deep SVDD and Deep SVDD ensemble methods underperform in all three metrics. The flow model alone is a very robust method, but the combination with the Deep SVDD ensemble allows one to obtain higher maximum TI scores.

On the secret set however the flow model underperforms in the minimum, median and maximum TI metrics when compared to the combined model. However, in this case the combined methods offer vast improvement.

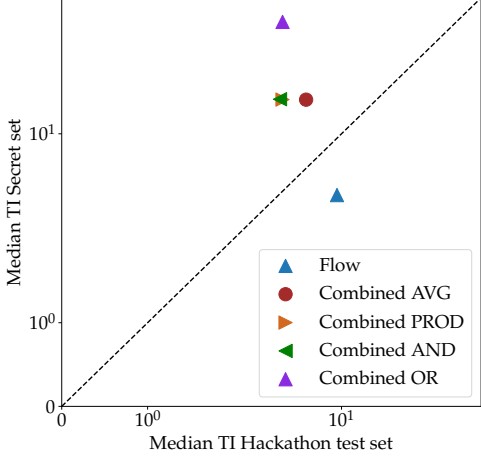

Figure 4: Comparison of the median TI between the hackathon test set and the secret set. While the performance of the flow model is worse on the secret set, the combined models do much better.

# 7 Conclusions

We show that a combination of an ensemble of Deep SVDD models, which targets *different* events, and a flow model, which targets *rare* events, can be used to robustly detect anomalies in high energy physics data. On the hackathon test set, the flow model yields the best result on the minimum and median TI scores, but the combined method has a significantly higher maximum TI score. However, on the secret set the combined models outperform the flow model in all three metrics. The OR combination algorithm yields the highest median and maximum TI, while the AVG combination algorithm leads to the highest minimum TI. We believe the combination algorithm outperforms the flow model alone because of the absence of training data in the tails of distributions. In such regions of low statistics the flow model tends to overestimate the likelihood (see e.g. [38]), and as such it may perform worse on extremely anomalous events. The ensemble of SVDD models should not be as sensitive to this issue, as any highly anomalous event should be mapped far away from the fixed target. As a result, the combination of the two methods leads to a more robust result when the anomalous events in the dataset are both *rare* or *different*, instead of mainly one or the other. We believe this method is effective in other data sets as well for detecting anomalies.

# 8 Discussion & Outlook

Anomaly detection using unsupervised models is gaining traction within the field of High Energy Physics, and rightfully so. It represents a very powerful method of searching for new physics. However, most methods focus on autoencoder models, like VAEs, $\beta$-VAEs, convolutional autoencoders or other variations. These only target events that are *different*, and might still reconstruct outliers well. The Deep SVDD method appears to be a better choice for detecting *different* events.

New physics might also arise by a surplus of *rare* events. This means that these events are actually present in the dataset, only in far fewer numbers than with a particular signal added on top of it. Because these events are present in the dataset, they fit the SM manifold and are therefore be assigned a low anomaly score by autoencoders or Deep SVDD models. This is typical for the above method of searching for anomalies: when ideally trained, all events will either be assigned a very low or a very high anomaly score. The flow model however assigns a likelihood given the probability density of the training set, meaning it provides more granular anomaly scores. Utilizing both algorithms gives a robust anomaly detector that will detect both *rare* and *different* events. Finally, while we develop this method specifically for detecting anomalies in high energy physics, we believe the method is useful for detecting anomalies in other datasets as well.

Finally, it should be noted that a mismodelling of the background could lead to a poor performance in two separate ways. It might lead to the misclassification of anomalous events. However, this can also be the case for many current LHC searches where one would use techniques such as BDTs, which are almost always trained on signal and background simulations, to define a signal region on which a statistical test can be applied to confirm or exclude a particular signal. In either case, this is usually not a big problem as the simulations are well validated and generally describes the LHC data within the known uncertainties.

Second, mismodelling can also lead to an underestimation of the background in the signal-enhanced selection, e.g. defined by a large signal or anomaly score. With our anomaly score method the same systematic variation can be performed as in the widely accepted signal-vs-background based searches, like the BDT approach: Monte Carlo background predictions can be changed, the training can be repeated with different systematics, background predic-

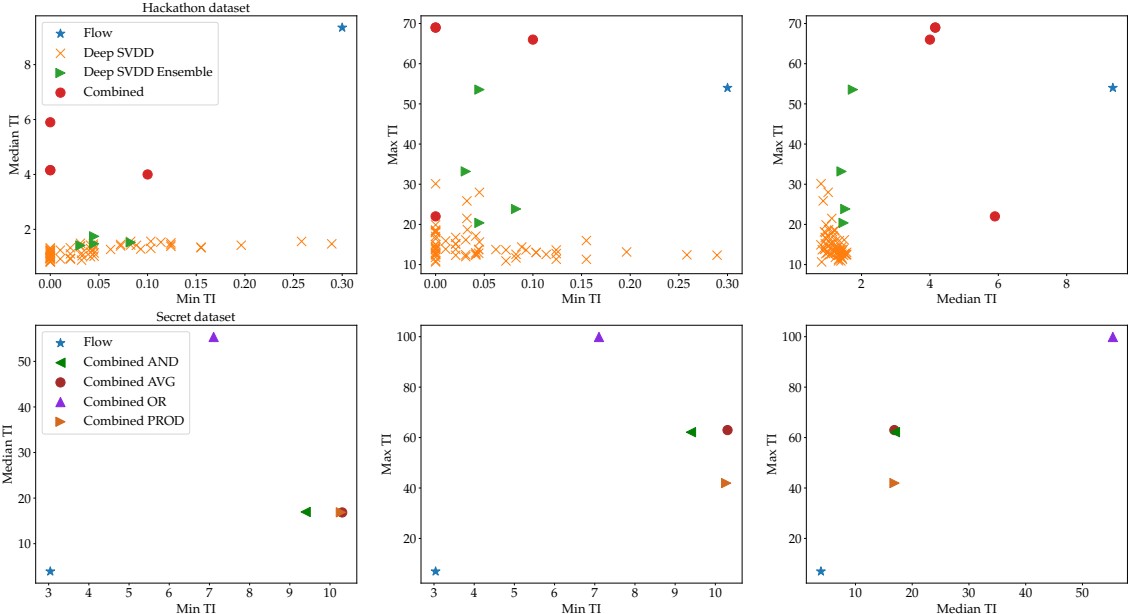

Figure 5: Minimum, median and maximum Total Improvement for the different models tested on the hackathon test set (top row) and the secret set (bottom row). On the secret set the combined model with the OR combination algorithm generally outperforms all other models.

tions can be constrained via control selections, etc. As such, we expect good stability of the anomaly score, as experiments like ATLAS and CMS typically see good stability in their signal-vs-background analyses.

# Acknowledgments

RV acknowledges support from the European Research Council (ERC) under the European Union's Horizon 2020 research and innovation programme (grant agreement No. 788223, PanScales), and from the Science and Technology Facilities Council (STFC) under the grant ST/P000274/1.

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
