# Peer review of "Rare and Different: Anomaly Scores from a combination of likelihood and out-of-distribution models to detect new physics at the LHC"

_SciPost Physics, doi:SciPost Phys. 12, 077 (2022)_

## Round 4 · Referee Report · Anonymous · 2021-9-20

Report

The authors study new approaches to anomaly detection in the context of the DarkMachines Anomaly Score challenge. They examine different combinations of One-Class Deep Support Vector Data Description models (for detecting out of sample anomalies) and autoregressive flows (for detecting rare in-sample anomalies). They quantify the performance of these combinations using the DarkMachines Anomaly Score datasets, using versions of the significance improvement (SI) score. Since the DarkMachines datasets contain many different signals, they consider the median, max and min significance improvements across all the different signals.

This work has some novel and interesting aspects and advances our understanding of a very important problem (model independent searches for new physics at the LHC). However, before it can be published, there are several questions about the methodology that I believe must be addressed:

- The Deep SVDD model seems like it could trivially learn the constant function, and then it would have no anomaly detection power. What prevents it from just mapping all inputs (regardless of signal or background) trivially to a constant? The authors should explain why this apparent, obvious failure mode does not happen for their Deep SVDD model.

- The models considered in this work were trained on a background-only sample, and then evaluated on background and various signals. If I understand correctly, for both training and evaluation the backgrounds were drawn from the same distribution (i.e. produced with the same generator and detector simulation). In that case, the anomaly scores being computed here could be very misleading. In particular, if the SM background in the data is not sufficiently well modeled by simulation, the sensitivity to new physics could be significantly worsened (it might flag the entire dataset as anomalous). The authors should discuss this issue at length and as quantitatively as possible. Why do they expect the SI metrics they computed here to be at all relevant if the background in the data is mismodeled by simulation?

- Anomaly scores are only one component of a successful new physics search strategy. Obviously, background estimation is another, equally important component. As far as I could tell, there was no mention of background estimation anywhere in this paper. The authors should include a discussion about how they imagine they could combine their anomaly score with an accurate method of background estimation.

---

## Round 5 · Referee Report · Anonymous · 2022-1-5

Report

Yes

---

## Round 5 · Author Response

Dear referee,

Thank you for reading our article and for the feedback to our submission. We have updated several paragraphs that incorporates the given feedback. Below our individual responses to the points:

point 1:
This is an excellent remark which is also discussed in the original Deep SVDD paper. The solution proposed there is to remove bias from the neurons and not use 0 as a target output value. However we found that the neural network never converged to this trivial solution. We have added a paragraph to the Deep SVDD section explaining the problem, solution and reasoning.

point 2:
We have added a discussion of this issue to the end of the discussion. In short, mismodelling of the background will indeed lead to worse performance, either through a misclassification of anomalies, or through failing subsequent statistical tests. We want to emphasise that the methods proposed here represent a very minor adjustment to the techniques the experiments currently use, only replacing the conventional signal scores by an anomaly score. As such, any negative effects due to mismodelling of the background would also apply to the current techniques.

point 3:
Background estimation is a powerful way to determine signal regions where we can perform counting experiments as well. However, our method is a way to detect anomalous using an event-by-event scoring system, as was the requirement of the Dark Machines challenge (2105.14027), for which our method is the overall winning one. These methods have the advantage that they allow for the use of the tools that are already widely available and used in the experiments. Anomaly detection that involves background estimation is in some sense an orthogonal approach, and we have added a paragraph in the introduction to clearly distinguish the two.

---

## Round 5 · List of Changes

- Added paragraph about dealing with trivial solutions for the Deep SVDD model (page 6)
- Added paragraph about misleading scores due to mismodeling of the data (page 12)
- Added paragraph about background estimation and how those methods relate to ours (page 2)

---

## Editorial Decision

published